# Mating suppresses sperm-dependent male avoidance in *C. elegans* hermaphrodites

Satoshi Suo [ID]*

Department of Pharmacology, Faculty of Medicine, Saitama Medical University, Saitama, Japan

* ssuo@saitama-med.ac.jp

## Abstract

In many sexually reproducing animals, females incur higher reproductive costs and therefore tend to be more selective in accepting mates. In *Caenorhabditis elegans*, self-fertilizing hermaphrodites produce a limited number of self-sperm, and previous studies have suggested that hermaphrodites avoid males. However, the behavioral dynamics of this male-avoidance behavior remain largely unexplored and its underlying mechanisms are not well-understood. Here, I quantitatively analyzed male-avoidance behavior in *C. elegans* hermaphrodites by measuring locomotor speed in the presence of males. Automated image analysis revealed that wildtype hermaphrodites increased speed when in contact with males, indicating active avoidance behavior. In contrast, avoidance was significantly reduced in sperm-deficient mutant hermaphrodites and aged hermaphrodites that had exhausted self-sperm. Similarly, females of gonochoristic *Caenorhabditis* species, which lack self-sperm, also showed no avoidance of males. These results suggest that the presence of self-sperm promotes male avoidance, likely to favor self-fertilization. Interestingly, hermaphrodites that had previously mated with males showed reduced male avoidance. Given that male-derived sperm outcompete self-sperm for fertilization, continued avoidance after mating may be no longer advantageous for reproduction. These findings highlight the adaptive nature of sperm-dependent male avoidance in *C. elegans* hermaphrodites.

## Introduction

In sexually reproducing animals, females tend to be more selective in accepting mates, as they typically incur higher reproductive costs. Female receptivity to males depends on their internal reproductive state, such as pregnancy status, sperm availability, or previous mating experience [1–4]. Although such modulation has been reported in species like *Drosophila* and mice, the underlying mechanisms are not fully understood. In *Caenorhabditis elegans*, hermaphrodites produce a limited amount of self-sperm during larval development, which is used for self-fertilization in adult animals [5]. Males search for hermaphrodites and attempt mating by locating the vulva

**Data availability statement:** All the relevant data and the codes used in this study are available on Figshare (https://doi.org/10.6084/m9.figshare.30577721).

**Funding:** This work was supported by JSPS KAKENHI Grant Number 19K06911 and 22K06322.

**Competing interests:** The authors have declared that no competing interests exist.

with their tail. Hermaphrodites are known to avoid males through increased locomotion [6–9]. Notably, hermaphrodites that exhausted self-sperm have been reported to exhibit reduced male avoidance [7]. Similarly, females of gonochoristic *Caenorhabditis* species, which lack self-sperm entirely, appear more receptive to males [10]. These findings suggest that male-avoidance behavior may be influenced by sperm status, but quantitative analyses remain limited.

In this study, I developed a method to quantitatively assess male-avoidance behavior in *C. elegans* hermaphrodites by measuring the speed of hermaphrodites when moving alone or in contact with males. Wildtype hermaphrodites increased speed upon contact with males, suggesting active avoidance. This response was diminished in sperm-less animals. These results support the idea that hermaphrodites prioritize using self-sperm and adjust their behavior to avoid mating when self-fertilization is possible. Interestingly, previously mated hermaphrodites no longer avoided males. Since male-derived sperm outcompetes self-sperm [11], further avoidance is unlikely to offer a reproductive advantage once mating has occurred. The results suggest that male-avoidance behavior is dynamically regulated based on reproductive context.

## Materials and methods

### Strains

Culturing of *C. elegans* was performed as previously described [12]. The strains used in this study are as follows: N2 wildtype (RRID:WB-STRAIN:N2), CB4088 *him-5(e1490)* (RRID:WB-STRAIN:CB4088), BA821 *spe-26(hc138)* (RRID:WB-STRAIN:BA821), EM464 *C. remanei* (RRID:WB-STRAIN:EM464), and CB5161 *C. brenneri* (RRID:WB-STRAIN:CB5161). CB4088 *him-5(e1490)* was used to obtain males for mating with *C. elegans* strains.

### Preparation of animals and image acquisition

Adult animals used for examining male-avoidance behavior were prepared as described [13,14] with some modifications. The assay plates were made of low peptone-NGM agar (0.25 g/L peptone, 3 g/L NaCl, 17 g/L agar, 25 mM KPO4 (pH 6.0), and 5 mM MgSO4, 5 mM CaCl2). 10 µL of OP50 (RRID:WB-STRAIN:OP50) suspension was placed on the plates and incubated at 22°C for 24 h to create thin and small bacterial lawns. The size of the bacterial lawns was approximately 10 mm.

Animals were cultured at 16°C. For the experiments with N2 and *spe-26*, L4 hermaphrodites were picked from the plates at 16°C and placed on a 40 mm NGM plate [12] seeded with OP50 and were grown without males at 22°C for approximately 24 h to obtain young adult hermaphrodites. The *spe-26* mutant is a temperature-sensitive sperm-defective mutant [15] and produced no viable progeny and only laid unfertilized eggs when cultured in this condition [14]. L4 males of him-5 mutants were also picked and cultured separately from hermaphrodites at 22°C for 24 h, to obtain adult males. *C. remanei* and *C. brenneri* were prepared in the same way except that females and wildtype males were used.

A hermaphrodite (or a female) and a male were transferred to the assay plate approximately 5 min prior to recording. The images of the assay plates were captured using a DMK series USB camera (Imaging Source, Bremen, Germany) with a resolution of 1,600 x 1,604 pixels, at one frame per sec for 60 min, using the gstreamer software with Raspberry Pi 3 (Raspberry Pi Foundation, Cambridge, United Kingdom).

To test male-specificity of the contact-induced acceleration, N2 hermaphrodites were placed with another N2 hermaphrodite (smaller young adults were used to distinguish two animals by size), and recorded.

For experiments with old hermaphrodites, L4 animals were cultured at 22°C for 72 h (3 days) and transferred to new plates every day. To verify sperm depletion, 1-day- and 3-day-old hermaphrodites were individually placed on NGM plates and allowed to lay eggs for 17 hours, after which the adults were removed. The number of viable progenies were counted 48 hours later.

When testing the effect of prior mating, 20 L4 hermaphrodites were placed on 40 mm NGM plates together with 20 L4 males and cultured for 24 h. For the control, unmated hermaphrodites were prepared by picking only L4 hermaphrodites and culturing for 24 h. *spe-26* hermaphrodites laid no fertilized eggs when they were cultured by themselves but laid numerous fertilized eggs when cultured with him-5 males, demonstrating that the males mated with hermaphrodites in this condition. The mated hermaphrodite (or female) was placed on an assay plate with an unmated male. Then, the images were captured for 60 min.

To confirm that this condition results in successful mating, L4 males were stained with MitoTracker Red CMXRos (ThermoFisher, Waltham, MA) before being placed with L4 hermaphrodites [14]. After co-culture on plates without Mitotracker, hermaphrodites were examined by fluorescent microscopy. Fluorescent signal was observed in the spermatheca region in 98% of the hermaphrodites (41/42) paired with males, whereas no fluorescence was detected in hermaphrodites cultured without males (0/42).

## Analyses of image data

The speed of animals was calculated using a custom-written code in Python. First, the area of the bacterial lawn was determined by thresholding one image for each recording that contains 3600 images, and it was confirmed by manually checking the image. Only the animals within the bacterial lawn were analyzed. Contours of animals were extracted by subtracting the background and thresholding, and the size and the position of the contours were determined. If there are two contours in one image, one must be that of a male and the other must be that of a hermaphrodite (female). Since males are smaller than hermaphrodites, the larger contour was categorized as that of the hermaphrodite, and the smaller contour was categorized as that of the male. This provides the range of the size for the male and the hermaphrodite, since there typically are many images (time points) with two contours in the image. If there is only one contour in an image, this could represent either a male, a hermaphrodite, or both animals in contact (an animal could be out of the bacterial lawn). This was determined by the size of the contour. If the size was within the range of the single male or hermaphrodite size (determined above), the contour was classified as a male or a hermaphrodite, respectively. If the size was larger than the range of the hermaphrodite, the contour was classified as the two animals in contact. After the classification, the speed of the animal was determined by averaging the distance moved between consecutive frames. This gives the speed of the male not in contact with the hermaphrodite, the hermaphrodite not in contact with the male, and the male and hermaphrodite in contact for each test. For each animal, the average values obtained across all recorded frames were used as a single data point in subsequent analyses.

The proportion of time spent in contact was also determined by dividing the time spent in contact by the time two animals were not in contact. Frames in which at least one of the animals was outside of the bacterial lawn were removed from the analysis.

To determine how locomotor speed changes around contact between hermaphrodites and males, I performed an event-aligned (peri-contact) analysis of speed. Tracking data were first segmented into contact bouts. Only bouts lasting at least

10 s were included. For each contact bout, locomotion speeds of hermaphrodites and males were extracted for a 10-s window before and after each event. To determine whether hermaphrodites or males were leading movement during physical contact, we analyzed contact periods with high locomotion speed. From the tracking data of N2 hermaphrodite-*him-5* male pairs, we extracted video frames corresponding to contact bouts where the speed exceeded 250 µm/s. For each high-speed contact segment, the corresponding video images were manually reviewed to identify which animal was positioned ahead and actively moving forward. Across 12 independent hermaphrodite–male pairs examined, the larger animal (i.e., the hermaphrodite) was leading in all 12 cases (100%).

### Statistical analysis

The number of animals tested for each experiment is shown in the figure legends. Each animal served as an independent biological replicate, contributing one summary value. Statistical analyses were performed using Python. Two-group comparisons were assessed using the Wilcoxon rank-sum test (two-tailed). For data sets with more than two groups, pairwise Wilcoxon rank-sum tests were performed with Bonferroni correction.

Effect sizes (rank-biserial correlation and Hodges–Lehmann median difference) and 95% confidence intervals were estimated by bootstrap resampling (2,000 iterations). Differences were considered statistically significant at \*$p < 0.05$, \*\*$p < 0.01$, \*\*\*$p < 0.001$. P values are shown in the figures, and other statistical values are summarized in S1 Table. For box plots, boxes indicate the interquartile range, and central lines represent the medians. Whiskers represent the most extreme values within 1.5 times the interquartile range.

## Results

### Quantification of male-avoidance behavior

It has been shown that *C. elegans* hermaphrodites have low locomotor activity in the presence of food, but when males contact them, they avoid males by increasing their speed [6–8]. To quantify this male avoidance behavior, a wildtype N2 hermaphrodite and a *him-5* male were placed on a plate with a small bacterial lawn and their locomotion was recorded for 60 min. Through background subtraction and thresholding, the contours of animals were classified as a hermaphrodite, a male, or a hermaphrodite and a male in contact, based on their size. Then, the average speed was determined for each of them (Fig 1A, B). The proportion of time males and hermaphrodites were in contact with each other was also determined (Fig 1C). It is known that a male makes contact with a hermaphrodite with its tail, moves around the hermaphrodite's body and follows the hermaphrodite if the hermaphrodite moves, until the male locates the vulva of the hermaphrodite or loses contact with the hermaphrodite. It is therefore likely that the speed of animals in contact is primarily determined by the hermaphrodites.

To test which animal drives the movement during contact, I visually examined video segments in which animals were in contact and moving rapidly (≥250 µm/s). In all 12 hermaphrodite–male pairs analyzed, the hermaphrodite was leading the movement (S3 Video).

The average speed of wildtype hermaphrodites was lower than that of animals in contact, indicating that hermaphrodites increase speed when in contact with males (Fig 1A; S2 Video). This result is consistent with previous studies [7] and demonstrates that the behavioral analysis developed here can quantify male-avoidance behavior of *C. elegans* hermaphrodites. Furthermore, males and hermaphrodites were in contact only for about 10% of the time (Fig 1C). This contact-induced acceleration was specific to cases where hermaphrodites were in contact with males, as animals did not exhibit increased locomotor activity when two hermaphrodites were in contact (Fig 1D), and the hermaphrodites stayed in contact only for a short amount of time (Fig 1E).

To further quantify locomotor dynamics around contact, I performed an event-aligned speed analysis (S1 Fig). Hermaphrodites showed a rapid acceleration immediately after contact onset, whereas males decelerated sharply, suggesting

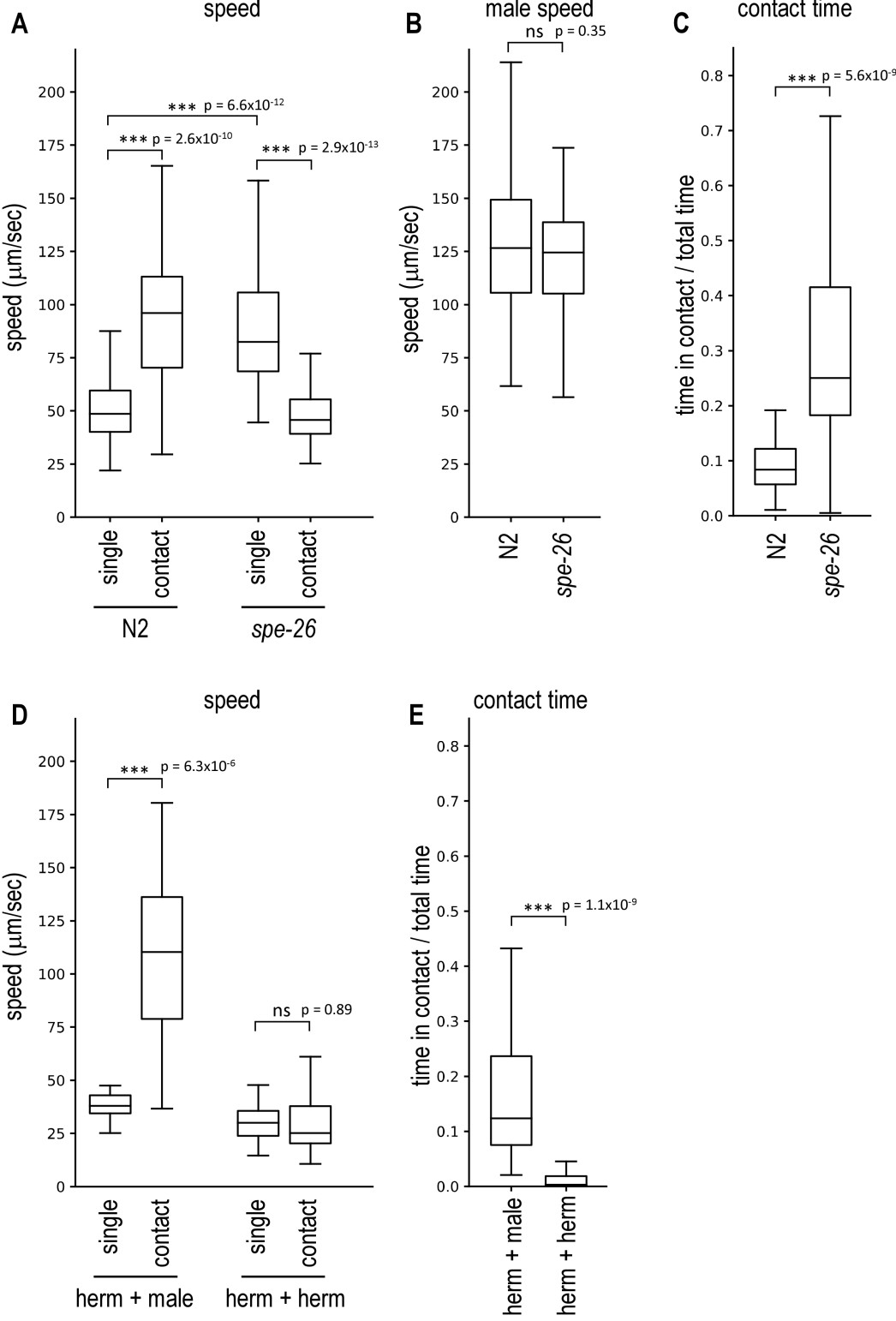

**Fig 1. Male avoidance of the wildtype N2 animals and the *spe-26* mutant.** A hermaphrodite and a *him-5* male were placed together on a plate and recorded **(A-C)**. **(A)** The speed of hermaphrodites not in contact with males (single) and of hermaphrodite–male pairs in contact (contact) was measured for the wildtype N2 animals and sperm-deficient *spe-26* mutants. **(B)** The speed of *him-5* males not in contact with hermaphrodites was determined. **(C)**

The proportion of time during which the hermaphrodite and the male were in contact was determined. Number of animals tested: N2, 59; *spe-26*, 52. N2 hermaphrodites were placed together with a *him-5* male or another N2 hermaphrodite and recorded to measure the speed (D) and the proportion of time in contact **(E)**. Number of animals tested: hermaphrodite + male, 20; hermaphrodite + hermaphrodite, 60.

that contact by males triggers acceleration of hermaphrodites. At contact offset, hermaphrodites increased speed while males slowed, suggesting that contact often ends because the hermaphrodite actively speeds up and breaks away.

   *C. elegans* hermaphrodites produce and store sperm during the larval stage and use the self-sperm to fertilize eggs in the adult stage and self-sperm has been implicated in male avoidance [7]. To test the effect of self-sperm, sperm-defective *spe-26* hermaphrodites were recorded together with *him-5* males. When *spe-26* mutant hermaphrodites are not in contact with males they move faster than wildtype hermaphrodites. This is consistent with the previous findings that sperm-deficient hermaphrodites exhibit increased locomotion [14]. When *spe-26* hermaphrodites were in contact with males, instead of speeding up as seen in wildtype hermaphrodites, the speed decreased (Fig 1A, S3 Video). This would be beneficial for males to locate the vulva. Furthermore, *spe-26* hermaphrodites spent more time in contact with males than wildtype hermaphrodites (Fig 1C). The speed of *him-5* males did not differ between when they were placed with N2 hermaphrodites and with *spe-26* hermaphrodites (Fig 1B). Taken together, the results show that sperm-deficient *spe-26* hermaphrodites exhibit reduced male avoidance and suggest that self-sperm (or resulting fertilized eggs) affect male-avoidance behavior.

## Old hermaphrodites exhibit reduced male avoidance

*C. elegans* hermaphrodites produce sperm in the larval stage and store it in the spermatheca. Mature hermaphrodites only produce eggs, and the stored self-sperm is used for fertilization. The self-sperm is known to be exhausted around 3 days after maturation and the old hermaphrodites, which keep producing eggs, require sperm from males to further produce fertilized eggs. It has been suggested that old hermaphrodites avoid males less as they require mating [7]. To confirm this, I also tested old animals. After 3 days, hermaphrodites laid only a few eggs ($1.6 \pm 0.5$ eggs in 17 hours), whereas 1-day hermaphrodites laid significantly more eggs ($142 \pm 8$ eggs, $p = 6.5 \times 10^{-6}$). Young adult hermaphrodites, which were cultured one day after the L4 stage, increased speed when in contact with males (Fig 2A). In contrast, old hermaphrodites, 3 days after the L4 stage, did not increase speed when in contact with males, although this change in speed was not sufficient to significantly increase the proportion of time in contact with males in old animals, compared to young animals (Fig 2B). Old *spe-26* hermaphrodites were also tested and continued to exhibit reduced male avoidance (Fig 2). The results demonstrate that male avoidance is reduced in old hermaphrodites after their self-sperm has been exhausted. The result is consistent with the previous study [7], further demonstrating the utility of this behavioral analysis.

## Male-female species exhibit no male avoidance

Previous studies has shown that females of gonochoric *Caenorhabditis* species do not exhibit male avoidance and are rather attracted to males unlike the hermaphrodites of closely related androdioecious species [10,16,17]. I tested *C. remanei* and *C. brenneri*, which are gonochoric and have females instead of hermaphrodites. Similar to sperm-deficient *spe-26* mutants (functionally females), *C. remanei* and *C. brenneri* females reduced speed when in contact with males and the proportion of time in contact with males was higher than the wildtype *C. elegans* hermaphrodites (Fig 3). The results further confirm that sperm-less animals exhibit reduced male avoidance.

## Mating suppresses male avoidance

The results suggest that the presence of self-sperm promotes male avoidance in hermaphrodites. This would allow hermaphrodites to preferentially use self-sperm over male-derived sperm. I next examined what effect male-derived sperm had on male-avoidance behavior in hermaphrodites. N2 hermaphrodites were cultured with *him-5* males for 24 hours

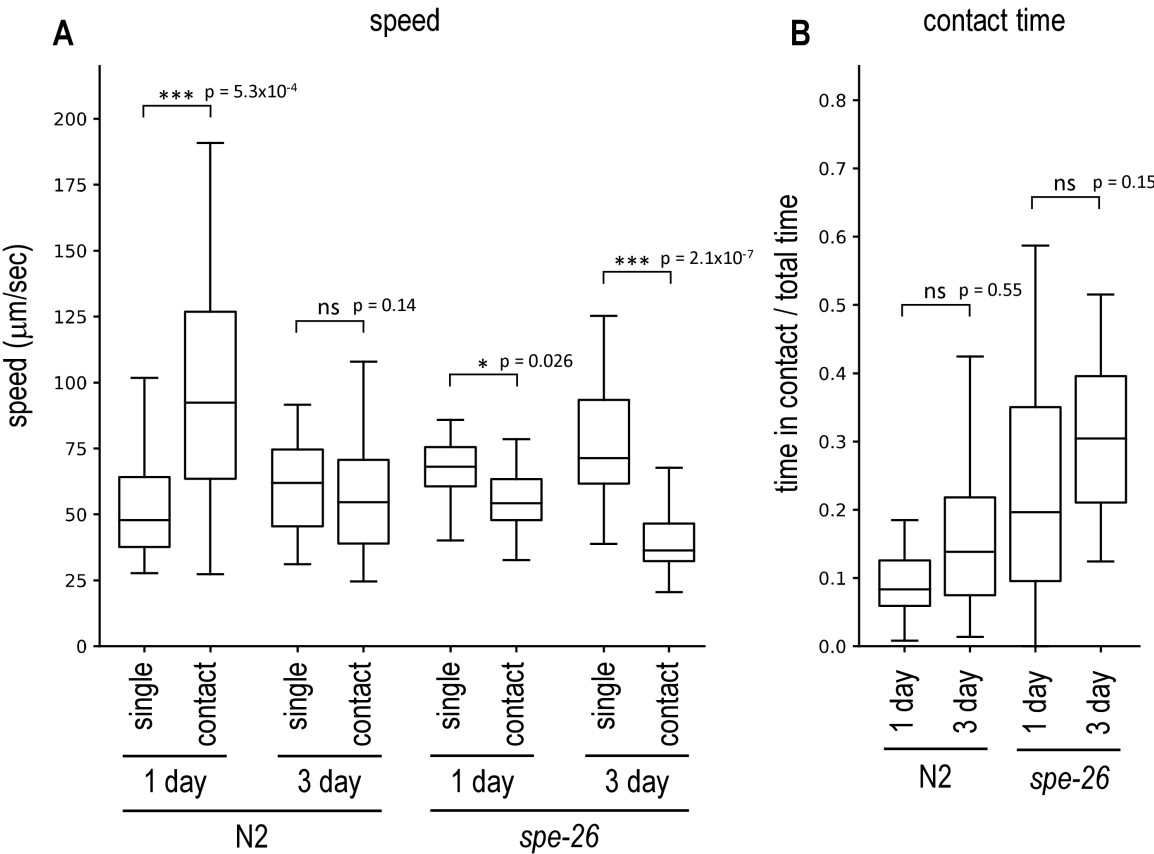

**Fig 2. Male avoidance of young and old hermaphrodites.** Wildtype hermaphrodites cultured for 1 day or 3 days after the L4 stage were placed on a plate with a *him-5* male and recorded. **(A)** The speed of hermaphrodites not in contact with males (single) and of hermaphrodite–male pairs in contact (contact) was measured for wildtype N2 and sperm-deficient *spe-26* mutants. **(B)** The proportion of time during which the hermaphrodite and the male were in contact was determined. Number of animals tested: N2 1 day, 32; N2 3 day, 32; *spe-26* 1 day, 31; *spe-26* 3 day, 32.

to mate with each other. (I confirmed that this condition results in 98% mating success by mating hermaphrodites with Mitotracker-labelled males.) Then, the hermaphrodite was placed on an assay plate with a virgin *him-5* male and their behavior was recorded. In wildtype hermaphrodites, mated hermaphrodites did not increase speed when in contact with a male, unlike the control unmated hermaphrodites, which increased speed when in contact with a male (Fig 4A), showing that mating suppresses male avoidance of hermaphrodites. However, mating did not significantly change the proportion of time animals were in contact (Fig 4B). I also tested sperm-deficient *spe-26* hermaphrodites. Both mated and unmated *spe-26* hermaphrodites exhibited reduced speed when in contact with males (Fig 4A). These results suggest that hermaphrodites with male-derived sperm have low levels of male avoidance.

## Discussion

In this study, I analyzed the male-avoidance behavior of *C. elegans* hermaphrodites. By measuring the speed of hermaphrodites moving alone and in contact with males, it was shown that hermaphrodites moved faster when in contact with males. These results suggest that hermaphrodites accelerate to avoid males. It was previously shown that old hermaphrodites that have exhausted sperm avoid males less [7]. Also, females of gonochoric *Caenorhabditis* species have been shown to slow down to accept males. Here, by measuring the speed of hermaphrodites and females, it was confirmed that these sperm-less animals do exhibit reduced male avoidance.

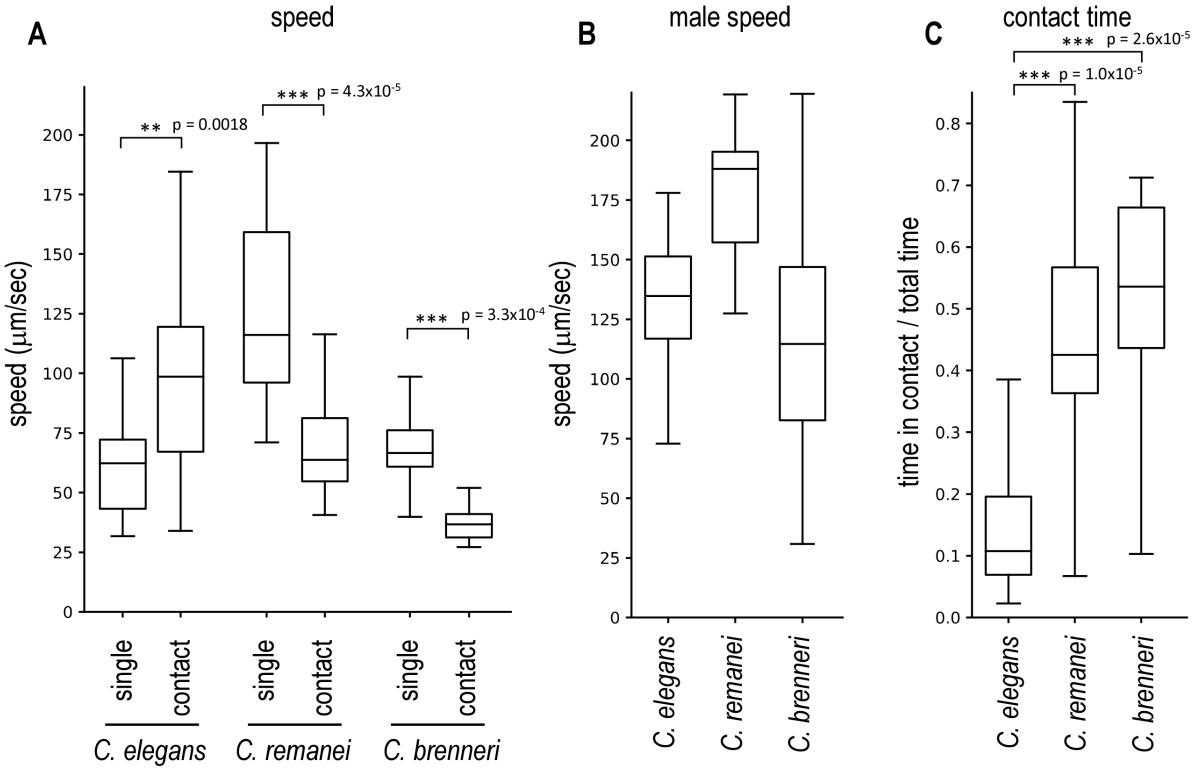

**Fig 3. Male avoidance of females of gonochoric species.** A female and a male of *C. remanei* or *C. brenneri* were placed together on a plate and recorded. **(A)** The speed of females not in contact with males (single) and of female–male pairs in contact (contact) was measured. *C. elegans* N2 hermaphrodites and *him-5* males were also tested. **(B)** The speed of males not in contact with females or hermaphrodites was determined. **(C)** The proportion of time during which the female (or hermaphrodite) and the male were in contact was determined. Number of animals tested: *C. elegans*, 37; *C. remanei*, 21; *C. brenneri*, 16.

These behavioral changes are likely to be beneficial for hermaphrodites in terms of transmitting their genes to the next generation. Wildtype *C. elegans* hermaphrodites have both sperm and eggs. Progeny from self-fertilization inherit 100% of their genes from the parent. If the hermaphrodite mates with a male, it is known that male-derived sperm is preferentially used for fertilization over self-sperm [11]. Therefore, most of the progenies have 50% of their genes inherited from the hermaphrodite. In terms of reproducing more of their own genes, self-fertilization is more beneficial for the hermaphrodite. Thus, as long as the hermaphrodite has sperm, it would be adaptive for her to avoid males. But when hermaphrodites have no self-sperm, either in old animals or sperm-deficient animals, avoiding males results in no progeny, which would be worse than producing cross-progeny that carry 50% of the genes from hermaphrodites. Therefore, it is no longer adaptive for hermaphrodites to avoid males, and as shown by the results, they did not increase speed when in contact with males. This was also true for females of gonochoric species.

I also found that hermaphrodites that were previously mated with males no longer avoid males. These mated hermaphrodites have both self-sperm and male-derived sperm, producing progeny that carry 50% of their genes from the hermaphrodite. Mating with another male does not change that only 50% of progeny's genes come from the hermaphrodite. Therefore, it no longer is beneficial for the mated hermaphrodite to avoid males, which may explain the observed behavioral change. The reduced male avoidance may also be advantageous by allowing hermaphrodites to acquire higher-quality sperm. It is demonstrated that sperm competition occurs between male-derived sperm in *C. elegans* [18,19]. Mating with another male

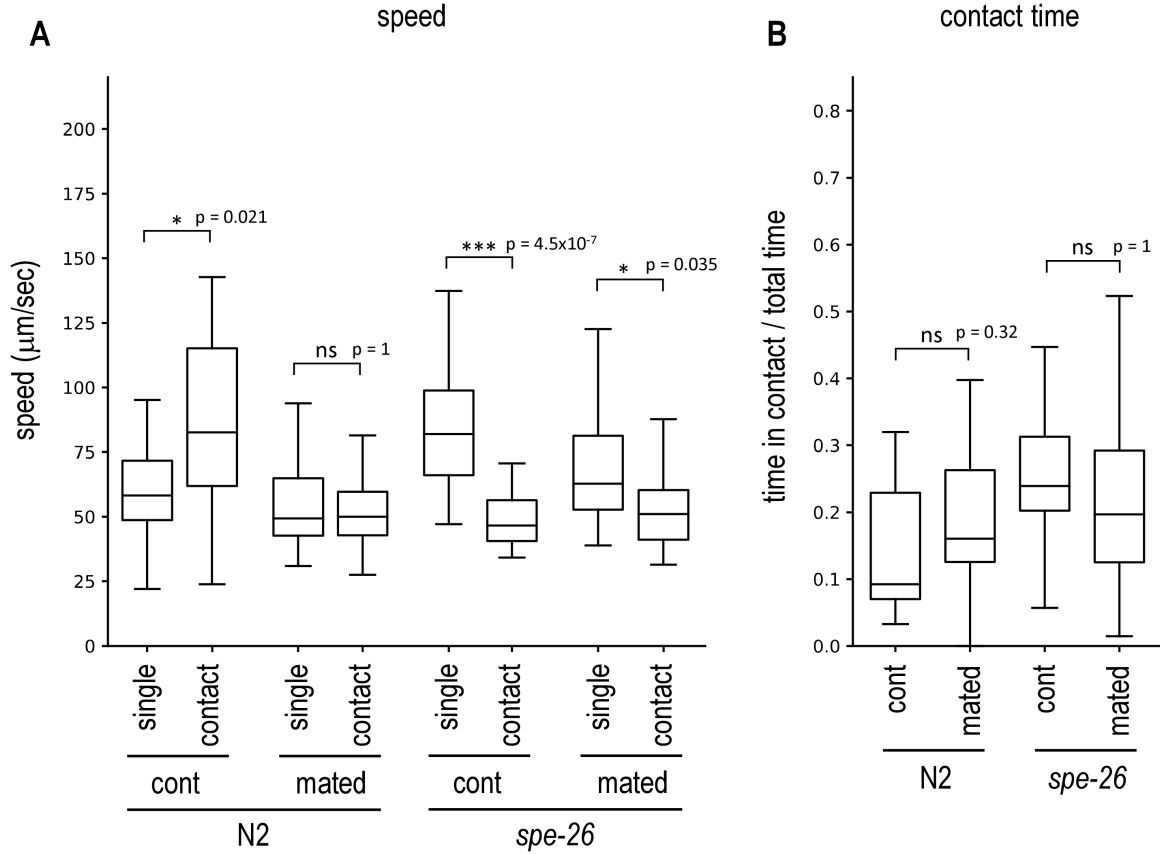

**Fig 4. Effect of mating on male avoidance. (A,B)** Hermaphrodites were cultured with *him-5* males for 24 hours. A mated hermaphrodite and a virgin *him-5* male were then placed together on a plate and recorded. **(A)** The speed of hermaphrodites not in contact with males (single) and of hermaphrodite–male pairs in contact (contact) was measured for wildtype N2 and sperm-deficient *spe-26* mutants. Comparisons were made between mated hermaphrodites (mated) and those not mated with males (cont). **(B)** The proportion of time during which the hermaphrodite and the male were in contact. Number of animals tested: cont N2, 35; mated N2, 35; cont *spe-26*, 34; mated *spe-26*, 35.

after first mating would make sperm from different males compete for fertilization and selects sperm, and therefore genes, that is better at fertilizing eggs.

The mechanisms by which self-sperm or mating with males alter male-avoidance behavior remain unknown. It was previously shown that *C. elegans* males are more active than hermaphrodites, presumably because males need to find mates to reproduce [13]. I also showed that sperm-deficient hermaphrodites, which also require mates to reproduce, exhibit increased locomotor activity [14], potentially allowing them to explore larger areas for mates. In this regulation, mating with males suppressed increased locomotor activity of sperm-deficient hermaphrodites. Therefore, male-derived sperm (or other stimuli from males) has the same effect as self-sperm in the regulation of locomotor activity. However, in the case of male avoidance, mating with a male has an opposite effect from self-sperm. In the future, it would be of interest to identify which male-derived stimuli induce the behavioral change and to elucidate the molecular and cellular mechanisms underlying this differential regulation.

In many animals, females choose whether to accept males depending on the internal state and the environment. By measuring the speed of hermaphrodites alone and in contact with males, this study confirmed that *C. elegans* hermaphrodites avoid males. And it was also shown that this male avoidance was decreased in hermaphrodites lacking self-sperm

and that mating with males suppressed male-avoidance behavior, suggesting the adaptive regulation of male avoidance by self-sperm and mating.

## Supporting information

**S1 Fig. Event-aligned locomotion speed around contact onset and offset.** Median locomotion speeds of hermaphrodites (red) and males (blue) aligned to contact onset (A) or offset (B). At contact onset, the hermaphrodite's speed increased ($p = 1.4 \times 10^{-5}$) while the male's speed decreased ($p = 6.9 \times 10^{-8}$). At contact offset, the hermaphrodite's speed again increased ($p = 0.0024$) while the male's speed decreased ($p = 5.8 \times 10^{-4}$). Shaded areas show the interquartile range and individual worms are shown as faint lines.
(TIF)

**S1 Video. Male follows a hermaphrodite during physical contact.** Representative video of an N2 hermaphrodite and a *him-5* male moving rapidly while maintaining contact (2x speed). Scale bar = 1 mm.
(AVI)

**S2 Video. Male avoidance behavior of N2 hermaphrodite.** An N2 hermaphrodite and a CB4088 male were recorded. The larger animal is the hermaphrodite. Only the moments when the animals are in proximity are shown (20x speed). Scale bar = 1 mm.
(AVI)

**S3 Video. Male avoidance behavior of *spe-26* hermaphrodite.** A *spe-26* hermaphrodite and a CB4088 male were recorded. The larger animal is the hermaphrodite. Only the moments when the animals are in proximity are shown (20x speed). Scale bar = 1 mm.
(AVI)

**S1 Table. Summary of statistical analysis.** P values, effect sizes and 95% confidence intervals obtained from the statistical tests are listed.
(XLSX)

## Acknowledgments

Some strains were provided by the Caenorhabditis Genetics Center. English editing was performed using ChatGPT (OpenAI) under the author's supervision.

## Author contributions

**Conceptualization:** Satoshi Suo.

**Data curation:** Satoshi Suo.

**Formal analysis:** Satoshi Suo.

**Funding acquisition:** Satoshi Suo.

**Investigation:** Satoshi Suo.

**Methodology:** Satoshi Suo.

**Project administration:** Satoshi Suo.

**Software:** Satoshi Suo.

**Writing – original draft:** Satoshi Suo.

**Writing – review & editing:** Satoshi Suo.

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
