## [Decision Letter · Decision Letter 0]

3 Oct 2025

Dear Dr. Suo,

Thank you for submitting your manuscript to PLOS ONE. After careful consideration, we feel that it has merit but does not fully meet PLOS ONE’s publication criteria as it currently stands. Therefore, we invite you to submit a revised version of the manuscript that addresses the points raised during the review process.

Both reviewers agree that the work described in the manuscript is sound and interesting. However, several comments from each reviewer must be addressed. Verbatim comments are attached at the end of this letter. Please carefully respond to their comments in a letter to reviewers and revise the manuscript accordingly. Look forward to receiving your revision.

We look forward to receiving your revised manuscript.

Kind regards,

Myeongwoo Lee, Ph.D.

Academic Editor

PLOS ONE

Journal Requirements:

“This work was supported by JSPS KAKENHI Grant Number 19K06911 and 22K06322.”

4. Please expand the acronym “JSPS KAKENHI” (as indicated in your financial disclosure) so that it states the name of your funders in full.

Reviewers' comments:

Reviewer's Responses to Questions

**Comments to the Author**

1. Is the manuscript technically sound, and do the data support the conclusions?

Reviewer #1: Yes

Reviewer #2: Partly

2. Has the statistical analysis been performed appropriately and rigorously?

Reviewer #1: Yes

Reviewer #2: I Don't Know

3. Have the authors made all data underlying the findings in their manuscript fully available?

Reviewer #1: Yes

Reviewer #2: Yes

4. Is the manuscript presented in an intelligible fashion and written in standard English?

Reviewer #1: Yes

Reviewer #2: Yes

Reviewer #1: The article “mating suppresses sperm-dependent male avoidance in C. elegans hermaphrodites” uses assays that measure locomotor speeds to investigate hermaphrodite C. elegans and C. remanei and C. brenneri behavior in the presence of males. The experiments in the article were done competently, and the results of the study are straightforward and consistent with published conclusions made by others in many other previous studies, i.e sperm-deficient/depleted females or hermaphrodites are more compliant to copulation than young hermaphrodites or females that contain sperm. The article has utility in that it describes a different assay to determine hermaphrodite mating receptibility; the work complements other assays that measure the same readout, mate receptiveness.

I have a small vanity comment concerning the section “Male-female species exhibit no male avoidance” . The behavior of virgin and impregnated C. remanei and C. brenneri females described in the section, have been previously addressed in the following two articles:

Chasnov JR, So WK, Chan CM, Chow KL. The species, sex, and stage specificity of a Caenorhabditis sex pheromone. Proc Natl Acad Sci USA 2007; 104:6730-5; PMID:17416682

and

Markert M, García LR. Virgin Caenorhabditis remanei females are attracted to a coital pheromone released by con-specific copulating males. Worm. 2013 Apr 1;2(2):e24448. doi: 10.4161/worm.24448. Epub 2013 Apr 18. PubMed PMID: 24058874; PubMed Central PMCID: PMC3704448.

The general conclusion made in this section has already been reported in these older articles, and thus should be acknowledged in addition to the Ebert 2024 article.

Reviewer #2: The study asks a clear behavioral question and the assay is tractable. You show that contact with males increases hermaphrodite speed and that this effect varies with sperm status, mating history, age, and species. The submission includes representative raw videos and usable summary tables, which is good. For acceptance, four points need to be resolved so that the central claim is supported by evidence that is both specific and reproducible.

First, attribution during contact needs to be demonstrated directly. Because the contact state merges two animals, the observed increase in speed could be driven by male propulsion rather than hermaphrodite avoidance. This can be addressed by an event-aligned analysis that plots hermaphrodite speed around contact onset and offset and shows acceleration at contact and relaxation after separation. A small hand-tracked subset where both centroids are followed through contact would also suffice. Either demonstration would make the causal interpretation secure.

Second, specificity must be shown with a non-mating conspecific control. The current contrasts imply male-specificity, but they do not exclude a general collision response. Pairing a hermaphrodite with a non-mating conspecific of comparable size, for example an adult hermaphrodite or an L4 hermaphrodite, and showing that contact with those animals does not elicit the same acceleration would resolve this. A small number of assays per condition would be adequate.

Third, statistical reporting and the unit of analysis need to match the design. The key comparisons are within animals across states. Please confirm the animal as the unit of analysis, ensure that each animal contributes a single summary value per state, use paired nonparametric tests for those contrasts, and report exact P values, effect sizes, and confidence intervals. If nonparametrics were chosen because normality or variance assumptions failed, state which tests were used and their results. Reporting the number and total duration of contact bouts per animal by condition will also help rule out biases from unequal bout structure.

Fourth, verification of biological states should be explicit. Because the interpretation turns on sperm status and mating history, the methods need to state how mating was confirmed in the wild-type animals that were later assayed and how sperm depletion was verified in the aged group, for example through cessation of fertilized egg laying under the reported conditions. Short, concrete criteria here will close a gap between the labels and the states being inferred.

If you add the attribution analysis, include the non-mating conspecific control, align the statistics to the within-animal design with complete reporting, and document verification of the biological states, the paper will meet PLOS ONE’s sound-science standard and will be acceptable.

**Do you want your identity to be public for this peer review?** For information about this choice, including consent withdrawal, please see our Privacy Policy

Reviewer #1: **Yes:** L. Rene Garcia

Reviewer #2: No

---

## [Author Response · Author response to Decision Letter 1]

10 Nov 2025

Response to Reviewers

I have carefully read the comments and have made changes that I hope meet with your approval. I provide a point-by-point discussion of the changes made to the manuscript.

The comments are also presented here by copying and pasting and my responses are indented. The “line” indicates where the changes were made in “Revised Manuscript with Track Changes.docx”. The code and relevant data are privately available on Figshare (https://figshare.com/s/e057de70396ce2e869e0) and will be made publicly available with a DOI (10.6084/m9.figshare.30577721) upon acceptance of the manuscript.

5. Review Comments to the Author

Reviewer #1: The article “mating suppresses sperm-dependent male avoidance in C. elegans hermaphrodites” uses assays that measure locomotor speeds to investigate hermaphrodite C. elegans and C. remanei and C. brenneri behavior in the presence of males. The experiments in the article were done competently, and the results of the study are straightforward and consistent with published conclusions made by others in many other previous studies, i.e sperm-deficient/depleted females or hermaphrodites are more compliant to copulation than young hermaphrodites or females that contain sperm. The article has utility in that it describes a different assay to determine hermaphrodite mating receptibility; the work complements other assays that measure the same readout, mate receptiveness.

I have a small vanity comment concerning the section “Male-female species exhibit no male avoidance” . The behavior of virgin and impregnated C. remanei and C. brenneri females described in the section, have been previously addressed in the following two articles:

Chasnov JR, So WK, Chan CM, Chow KL. The species, sex, and stage specificity of a Caenorhabditis sex pheromone. Proc Natl Acad Sci USA 2007; 104:6730-5; PMID:17416682

and

Markert M, García LR. Virgin Caenorhabditis remanei females are attracted to a coital pheromone released by con-specific copulating males. Worm. 2013 Apr 1;2(2):e24448. doi: 10.4161/worm.24448. Epub 2013 Apr 18. PubMed PMID: 24058874; PubMed Central PMCID: PMC3704448.

The general conclusion made in this section has already been reported in these older articles, and thus should be acknowledged in addition to the Ebert 2024 article.

REPLY: I have cited the two papers in line 223-225 as reference 16 and 17 (line 339-343). Thank you for pointing out.

Reviewer #2: The study asks a clear behavioral question and the assay is tractable. You show that contact with males increases hermaphrodite speed and that this effect varies with sperm status, mating history, age, and species. The submission includes representative raw videos and usable summary tables, which is good. For acceptance, four points need to be resolved so that the central claim is supported by evidence that is both specific and reproducible.

First, attribution during contact needs to be demonstrated directly. Because the contact state merges two animals, the observed increase in speed could be driven by male propulsion rather than hermaphrodite avoidance. This can be addressed by an event-aligned analysis that plots hermaphrodite speed around contact onset and offset and shows acceleration at contact and relaxation after separation. A small hand-tracked subset where both centroids are followed through contact would also suffice. Either demonstration would make the causal interpretation secure.

REPLY: As suggested, I performed an event-aligned analysis to directly examine locomotor changes around contact onset and offset. The methods have been added in Lines 130–133, and the results are shown in S1 Fig and described in Lines 184–188. At contact onset, the hermaphrodite’s speed increased while the male’s speed decreased, suggesting that contact initiated by the male triggers acceleration of the hermaphrodite. At contact offset, the hermaphrodite’s speed again increased while the male’s speed decreased, suggesting that contact often ends because acceleration of the hermaphrodite separates the male.

In addition, to determine which animal leads movement during contact, I visually examined video segments in which pairs were in contact and moving rapidly. In all observed contact events, the hermaphrodite was leading the movement, with the male following. The representative video has been added as S1 Video, with methodological details provided in Lines 134–140 and results in Lines 171–173.

Together, these analyses demonstrate that during contact, the hermaphrodite drives the movement, supporting that the observed acceleration can be attributed to hermaphrodite avoidance.

Second, specificity must be shown with a non-mating conspecific control. The current contrasts imply male-specificity, but they do not exclude a general collision response. Pairing a hermaphrodite with a non-mating conspecific of comparable size, for example an adult hermaphrodite or an L4 hermaphrodite, and showing that contact with those animals does not elicit the same acceleration would resolve this. A small number of assays per condition would be adequate.

REPLY: To test whether the observed acceleration is specific to contact with males, N2 hermaphrodites were paired with another N2 hermaphrodite (smaller young adults were used to distinguish the two individuals). The methods have been added in Lines 86–88, and the results are described in Lines 180–183. The results showed that contact between two hermaphrodites did not induce acceleration and the contact duration was short. The results are presented in Fig 1D and 1E, showing that the contact-induced acceleration occurs specifically during contact with males.

Third, statistical reporting and the unit of analysis need to match the design. The key comparisons are within animals across states. Please confirm the animal as the unit of analysis, ensure that each animal contributes a single summary value per state, use paired nonparametric tests for those contrasts, and report exact P values, effect sizes, and confidence intervals. If nonparametrics were chosen because normality or variance assumptions failed, state which tests were used and their results. Reporting the number and total duration of contact bouts per animal by condition will also help rule out biases from unequal bout structure.

REPLY: As suggested, I have clarified that each individual animal served as an independent unit of analysis, contributing a single summary value per condition (stated in Lines 125–126, Lines 143-144).

Nonparametric statistics were used since locomotor speed distributions are typically non-normal. I used unpaired Wilcoxon rank-sum tests because the experiments were not performed in a paired design. For multiple group comparisons, pairwise Wilcoxon rank-sum tests with Bonferroni correction were applied. I reported exact P values, effect sizes (rank-biserial correlation and Hodges–Lehmann median difference), and 95% confidence intervals. The exact P values are shown in the figures, and all statistical values are summarized in S1 Table. The methods for statistical analysis are described in Lines 143–152, and all analysis codes are available on Figshare.

Regarding contact structure, the total duration of contact varied among conditions. For example, contact duration was longer in sperm-deficient spe-26 hermaphrodites than in wild-type N2 animals (Fig 1C). This difference likely reflects the biological phenomenon under investigation, as reduced male avoidance would lead to prolonged contact.

Fourth, verification of biological states should be explicit. Because the interpretation turns on sperm status and mating history, the methods need to state how mating was confirmed in the wild-type animals that were later assayed and how sperm depletion was verified in the aged group, for example through cessation of fertilized egg laying under the reported conditions. Short, concrete criteria here will close a gap between the labels and the states being inferred.

REPLY: To verify successful mating, I stained L4 males with MitoTracker before pairing them with L4 hermaphrodites. After 24-hour co-culture, fluorescence was detected in the spermatheca region of 98% of hermaphrodites (41/42), confirming sperm transfer under this condition. The method is described in Lines 100–105 and the results are described in Lines 235-237.

To verify sperm depletion in old hermaphrodites, I quantified progeny production. Hermaphrodites cultured for 1 day after the L4 stage produced an average of 142 ± 8 progenies within 17 hours, whereas 3-day-old hermaphrodites produced only 1.6 ± 0.5 progenies, confirming the cessation of fertilized egg production most likely due to sperm exhaustion. The procedure is described in Lines 89–92 and the results in Lines 209-211.

These results are consistent with the biological states (mated and sperm-depleted) being as intended.

If you add the attribution analysis, include the non-mating conspecific control, align the statistics to the within-animal design with complete reporting, and document verification of the biological states, the paper will meet PLOS ONE’s sound-science standard and will be acceptable.

I am grateful for the reviewers’ valuable comments, which have greatly improved the manuscript. I hope that the revisions have adequately addressed all concerns and that the revised manuscript is now suitable for publication.

---

## [Decision Letter · Decision Letter 1]

15 Dec 2025

Mating suppresses sperm-dependent male avoidance in C. elegans hermaphrodites

PONE-D-25-47872R1

Dear Dr. Suo,

We’re pleased to inform you that your manuscript has been judged scientifically suitable for publication and will be formally accepted for publication once it meets all outstanding technical requirements.

Kind regards,

Myeongwoo Lee, Ph.D.

Academic Editor

PLOS One

Additional Editor Comments (optional):

Reviewers' comments:

Reviewer's Responses to Questions

**Comments to the Author**

Reviewer #1: All comments have been addressed

Reviewer #3: All comments have been addressed

2. Is the manuscript technically sound, and do the data support the conclusions?

Reviewer #1: Yes

Reviewer #3: Yes

3. Has the statistical analysis been performed appropriately and rigorously?

Reviewer #1: Yes

Reviewer #3: Yes

4. Have the authors made all data underlying the findings in their manuscript fully available?

Reviewer #1: Yes

Reviewer #3: Yes

5. Is the manuscript presented in an intelligible fashion and written in standard English?

Reviewer #1: Yes

Reviewer #3: Yes

Reviewer #1: The author has sufficiently addressed my suggestions concerning the section “Male-female species exhibit no male avoidance”.

Reviewer #3: (No Response)

**Do you want your identity to be public for this peer review?** For information about this choice, including consent withdrawal, please see our Privacy Policy

Reviewer #1: **Yes:** L. Rene Garcia

Reviewer #3: No

---

## [Editor Report · Acceptance letter]

PONE-D-25-47872R1

PLOS One

Dear Dr. Suo,

I'm pleased to inform you that your manuscript has been deemed suitable for publication in PLOS One. Congratulations! Your manuscript is now being handed over to our production team.

Kind regards,

on behalf of

Dr. Myeongwoo Lee

Academic Editor

PLOS One